# Preparation of Hydrophobic Surface on PLA and ABS by Fused Deposition Modeling

**DOI:** 10.3390/polym12071539

**Published:** 2020-07-12

**Authors:** Huadong Yang, Fengchao Ji, Zhen Li, Shuai Tao

**Affiliations:** Department of Mechanical Engineering, North China Electric Power University, Baoding 071003, China; jfc15735182916@163.com (F.J.); wllizhen@163.com (Z.L.); t746434208@163.com (S.T.)

**Keywords:** hydrophobic surface, fused deposition modeling, surface roughness, orthogonal experiment

## Abstract

In the fields of agriculture, medical treatment, food, and packaging, polymers are required to have the characteristics of self-cleaning, anti-icing, and anti-corrosion. The traditional preparation method of hydrophobic coatings is costly and the process is complex, which has special requirements on the surface of the part. In this study, fused deposition modeling (FDM) 3D printing technology with design and processing flexibility was applied to the preparation of hydrophobic coatings on polylactic acid (PLA) and acrylonitrile butadiene styrene (ABS) parts, and the relationship between the printing process parameters and the surface roughness and wettability of the printed test parts was discussed. The experimental results show that the layer thickness and filling method have a significant effect on the surface roughness of the 3D-printed parts, while the printing speed has no effect on the surface roughness. The orthogonal experiment analysis method was used to perform the wettability experiment analysis, and the optimal preparation process parameters were found to be a layer thickness of 0.25 mm, the Grid filling method, and a printing speed of 150 mm/s.

## 1. Introduction

Polylactic acid (PLA) is a polymer obtained by polymerization of lactic acid as the main raw material. Lactic acid is derived from raw materials such as corn and cassava. It has sufficient sources and is renewable. The production process of polylactic acid is pollution-free, and the product can be biodegraded to achieve circulation in nature, so it is an ideal green polymer material. Acrylonitrile butadiene styrene (ABS) is a terpolymer of acrylonitrile (A), butadiene (B), and styrene (S). The relative content of the three monomers can be arbitrarily changed to make various resins. ABS has the common properties of three components: A makes it resistant to chemical corrosion and heat, and has a certain surface hardness; B makes it have high elasticity and toughness; S makes it have the characteristics of thermoplastic molding and improves electrical properties. Therefore, ABS is a material with easy access to raw materials, good comprehensive performance, low price, and a wide range of uses.

ABS has been widely used in manufacturing industries such as the machinery, electrical, textile, automobile, aircraft, ship, and chemical industries. Polylactic acid (PLA) is widely used in agriculture, forestry, medical, and food industries, as well as packaging and other fields. Unique functions such as cleaning, anti-icing, anti-corrosion, water-reducing oil separation, liquid transport, and water collection [1,2,3,4,5,6] are needed in these fields. In daily life, the surfaces of automobiles, home appliances, and packaging with anti-fouling properties, as well as kitchen appliances with anti-fouling properties, are expected. For wind turbines and the power grid, at low temperatures, ice coating will occur on the surface of transmission lines, insulators, and wind turbines, which affect their performance. Therefore, the ability to resist ice coating is required in these fields [7]. The realization of these functions can be solved by constructing a hydrophobic layer, as wettability is a dominant factor for ice formation on a specific surface. Hydrophobic surfaces can tend to repel water [8]; however, hydrophilic surfaces have more affinity to water [9]. These functions can be obtained by coating a hydrophobic film on PLA and ABS surfaces.

Hydrophobic surfaces can be fabricated by modifying low-surface-energy materials on rough surfaces or constructing micro-nano rough structures on surfaces with low surface energy [10,11,12,13,14,15,16,17,18,19]. Many methods have been developed for the preparation of hydrophobic coatings, such as electrochemical deposition, photolithography, chemical etching, phase separation, spraying, sol–gel, self-assembly, and electro-spinning. However, these preparation methods of hydrophobic surfaces are complicated and costly.

3D printing is an emerging additive manufacturing technology that is developing rapidly [20], and has been widely used in rapid prototyping and customized products. Among the 3D printers on the market, fused deposition modeling (FDM) printers are popular because of their open source and simple system configuration, which have been widely used in the polymers field. In addition, a variety of materials can be applied for FDM printing, such as acrylonitrile butadiene styrene (ABS), polylactic acid (PLA), thermoplastic polyurethane (TPU), high-density polystyrene (HIPS), and polyamide (nylon) [21,22,23,24,25]. Studies have shown that the surface properties of parts affect their wettability, and the surface quality is affected by process parameters of FDM 3D-printed parts [26,27,28,29,30,31,32]. Jiang [33] analyzed the FDM forming process by a numerical model and experiment, and obtained the mechanism of dimensional error, and the function of specific surface accuracy error was constructed. Xue [34] analyzed the effect of layer thickness on forming quality. The results showed that if the layer thickness was smaller, excellent forming quality will be obtained; however, it will reduce the printing speed. Feng [35] studied the problems of nozzle clogging and substrate deformation that affected the dimensional accuracy during the 3D printing. The results show that the selection of high-quality materials and the use of optimized process parameters have a significant effect.

Based on the construction of the process parameters and objective function of fused deposition modeling, the application of artificial neural networks and genetic algorithms for process optimization of fused deposition modeling was discussed. According to the idea of converting the multi-objective optimization problem into the single-objective, the optimal single-objective was decomposed and the precision and warpage of the part under the optimal condition were obtained [36]. Based on the fused deposition modeling (FDM) technology, the influence of building orientation on surface roughness and supporting area is studied. The surface roughness of parts made by polylactic acid (PLA) at different building orientations is tested in experiments, and combined with the theoretical and empirical formula, a novel model for surface roughness prediction is established, which conforms to the actual situation in manufacturing [37].

In the current research, much research has been implemented to analyze the effect of the layer thickness and building orientation on the forming quality, but the research on the influence of the filling method on surface quality and mechanical properties is not enough, and further discussion is needed. The preparation of coatings on the surface of printed parts also requires in-depth research to explore the relationship between process parameters and surface wettability.

In this paper, a commercial FDM 3D printer was applied to print parts by using PLA and ABS filaments. By modifying the printing process parameters, multiple sets of test pieces were prepared, and the relationship between the process parameters and surface roughness of the test pieces was explored. Then, the preparation method of hydrophobic coatings on FDM test pieces was studied. Finally, the corresponding relationship between process parameters and surface hydrophobic properties was discussed. The surface quality and the surface roughness were observed and measured by using a light-cut microscope, and the effect of different printing parameters on the surface quality of the tested part was analyzed and optimized. Then, hydrophobic GF-2200 self-drying nano-ceramic fluorine paint was used to achieve a nano-structured hydrophobic coating surface. The contact angle measuring instrument was used to measure the static contact angle of the surface coating, and the wettability changes of the 3D-printed parts under different process parameters were evaluated and analyzed. The effect of different process parameters on the hydrophobic properties of the coatings on the 3D-printed part was studied.

## 2. Materials and Methods

### 2.1. Sample Preparation

In this experiment, tested parts were manufactured by an FDM printer (Liuyunkeji, Taian, China), which is shown in Figure 1a. The process principle of the FDM printer is shown in Figure 1b. The thermoplastic filament with a diameter of 1.75 mm is fed into the extrusion nozzle through the filament feeding roller. The thermoplastic filament is heated to the melting point. The hot, liquid plastic emerges through a nozzle to create a thin, single layer along the X- and Y-axes on the build platform. The layer cools and hardens quickly. As each layer is completed, the platform is lowered and additional molten plastic is deposited, growing the part vertically (along the Z-axis).

In order to prepare the hydrophobic surface, a three-dimensional grid-like solid model was designed by Pro/E software which is shown in Figure 2. The sample size was 30 mm × 30 mm × 3 mm, the size of the holes was 1.5 mm × 1.5 mm, and the distance between the holes was 1.5 mm. Simplify3D software was used to edit parameters and slice the solid model in the STL file, which was imported into an FDM printer. PLA and ABS material (detailed specifications are shown in Table 1 and Table 2) were applied to print a test part in order to evaluate the effect of materials on surface roughness and wettability. In this experiment, the layer thickness was set as 0.15, 0.2, and 0.25 mm, the printing speed was chosen as 100 and 150 mm/s, respectively, and the filling methods are shown in Figure 3, such as Rectilinear, Grid, Wiggle, and Honeycomb. For other process variables, the default parameters were chosen. In order to verify the effect of the printing process parameters on the surface roughness, 72 PLA test pieces and 72 ABS test pieces were printed by considering the three factors of printing speed, layer thickness, and filling mode.

### 2.2. The Measurement Method of Surface Roughness

The surface roughness of the samples that were manufactured by the FDM printer with different process parameters is characterized by measuring the micro-roughness *R*z with a light-section microscope. The detailed measurement steps are as follows:

(1) A 7 × lens was selected to install on the microscope, and the beam was focused on the object to be measured by coarsely adjusting the hand wheel. When a clear slit image appears in the eyepiece, the coarse adjustment hand wheel should be fixed and the fine adjustment hand wheel should be applied to adjust the sharpness of the slit image.

(2) Turn the side micro-eyepiece so the crosshairs are in the horizontal position, and then adjust the horizontal lines of the crosshairs to be tangent to the highest point of the clear edge of the slit, and then the number on the eyepiece reticle is recorded as $a_{1}$. Then, make the horizontal line of the reticle tangent to the lowest point of the clear edge of the slit, and the number on the eyepiece reticle is recorded as $a_{2}$.

(3) Repeat step (2) 5 times so the number of the five highest peaks and the five lowest valleys of the measured profile can be obtained. Then, the average value can be calculated by Equation (1). Thus, the surface roughness of the tested parts can be obtained by Equation (2).
(1)$$\;a_{average}=\frac{\left(a_{1}+a_{3}+\cdots +a_{9}\right)-\left(a_{2}+a_{4}+\cdots +a_{10}\right)}{5}\;$$
(2)$$\;R_{Z}=\frac{a_{average}}{2V}\;$$

(4) Determination of *V* value. The surface roughness of the scale can be measured by using a 0.01 mm standard roughness scale. The values of the five peaks and five valleys measured in steps (1), (2), and (3) are shown in Table 3.

According to the above equation, the values of *a* and *R*z can be calculated. As the *R*z of the standard roughness scale is 0.01 mm, the *V* value (6.3) can be computed by the following formula.
$$0.01=\frac{1.26}{2V}$$

According to the above methods and steps, the values of the tested parts are measured in sequence, and the roughness values *R*z can be calculated.

### 2.3. Hydrophobic Surfaces Preparation

A low-cost GF-2200 hydrophobic coating (GUANGFU, Guangzhou, China), which is a room-temperature curable single-component nano-ceramic resin with a maximum hydrophobic angle of 110°, was applied in the experiment. The preparation of hydrophobic surfaces on FDM tested parts is shown in Figure 4.

The surface of tested parts was cleaned to remove dust, oil, and other contaminants from the surface, and then dried. After completely drying, the GF-2200 self-drying nano-ceramic fluorine coating was dipped with 3D-printed parts for 1 min and was then dried at room temperature for 10 h. The dried dip-coated 3D-printed parts were immersed in ethanol and cleaned using an ultrasonic cleaner for 10 min.

### 2.4. Measurement of Coating Wettability of Tested Parts

Wettability refers to the conversion of a solid interface from a solid–gas interface to a solid–liquid interface when a liquid is in contact with a solid surface, which reflects the ability of the liquid to expand on the solid surface. The degree of wetting is expressed by the contact angle, which is defined as the angle between the tangent at the gas–liquid interface and the solid–liquid boundary at the gas–liquid–solid three-phase intersection, which is shown in Figure 5. When the droplet is stable on the solid surface, the tension at the gas–solid, gas–liquid, and solid–liquid interfaces reaches equilibrium.

Regarding the theoretical model of contact angle, Young proposed Young’s equation in 1805 [25]:$$\gamma_{LG}cos\theta =\gamma_{SG}-\gamma_{SL}$$
where $\gamma_{LG},\;\gamma_{SG},\;\gamma_{SL}$ represent the surface tension of the solid–gas, solid–liquid, and gas–liquid interface, respectively. θ is the balanced contact angle, or the intrinsic contact angle of the material. Young’s equation is only applicable to ideal smooth surfaces, and the accuracy of determining the wettability of rough surfaces in practice is reduced. In 1936, Wenzel proposed the effect of surface roughness on the wettability of the surface, and modified Young’s equation [38]:$$cos\theta_{w}=rcos\theta$$
where r is the surface roughness and $\theta_{w}$ is the contact angle of the rough surface in the Wenzel state. The Wenzel equation is only applicable to the thermodynamic equilibrium and stable state, but due to the uneven surface, the expansion of the liquid on the surface needs to overcome a series of potential barriers caused by unevenness. When the vibrational energy of the droplet is less than this potential barrier, the droplet cannot reach the equilibrium state required by the Wenzel equation and may be out of some metastable equilibrium state. In 1944, Cassie and Baxter [39] expanded the Wenzel model, and it was proposed that a rough and uneven solid surface can be imagined as a composite plane. Assuming that the solid surface is composed of two substances, their intrinsic contact angles are denoted by $\theta_{1}$ and $\theta_{2}$, respectively. The fractions of surface area occupied per unit area are $f_{1}$ and $f_{2}$, respectively. $\theta_{CB}$ is the intrinsic contact angle of the surface. The Cassie–Baxter model was established as shown in Figure 6b.

$$cos\theta_{CB}=f_{1}cos\theta_{1}+f_{2}cos\theta_{2}$$

The wettability of the samples is characterized by the value of contact angle. In this experiment, the images of a water droplet on different positions of samples can be obtained by a high-speed camera (Qianyanlang 2F04, Hefei, China), which is shown in Figure 7. The contact angle was measured based on the images (in bmp or JPG format) captured from the camera by contact angle measurement software, which is shown in Figure 8. Multiple measurements are averaged to eliminate measurement errors.

## 3. Results and Discussion

### 3.1. The Effect of Process Parameters on Surface Roughness

Many studies have shown that wettability has an important relationship with surface roughness, and the surface quality of FDM-printed parts is related to process parameters and materials. Therefore, this experiment is first aimed at the preparation of tested parts according to the following process parameters: Two materials (PLA and ABS), three layer thicknesses (0.15, 0.20, and 0.25 mm), four filling methods (Rectilinear R, Grid G, Wiggle W, and Honeycomb H), and two printing speeds (100 and 150 mm/s), as shown in Table 4. If three test pieces were printed for each parameter combination, a total of 72 PLA and 72 ABS tested parts needed to be fabricated for surface roughness measurement before coating preparation.

Existing research has found that the surface roughness of the substrate has a very significant effect on the performance of the hydrophobic coating. It can be found from Figure 9 and Figure 10 that the surface roughness of 3D-printed parts is more sensitive to the filling method and layer thickness, while the effect of printing speed and materials on the surface roughness is not obvious. It can be found from the figures that the surface roughness increases with the increase in layer thickness when other parameters remain unchanged. When the other parameters remain unchanged, the surface roughness of the grid filling method is the largest.

Because the preparation of the hydrophobic coating takes a long time, in order to reduce the number of experiments, an orthogonal test method was used for experimental design of the contact angle measurement. According to the above roughness experimental analysis results, the effectiveness of the orthogonal test method is verified. The orthogonal test table used is as shown in Table 5. 

From Table 6 and Figure 11, it can be found that the sensitive factors for surface roughness in FDM printing are layer thickness and filling mode. The result is consistent with the actual experimental results, which can demonstrate that the orthogonal experiment is effective and valuable.

### 3.2. The Effect of Process Parameters on Wettability

In order to construct a surface coating with optimal hydrophobic properties, an orthogonal test method was used to establish an orthogonal test ($L_{9}\left(3^{4}\right)$, as shown in Table 7) with three factors and three levels. According to the coating preparation method and contact angle measurement method mentioned in Section 2.3 and Section 2.4, the hydrophobicity evaluation test was performed on the test pieces printed by ABS and PLA materials, respectively. According to the established orthogonal test table, each material only needs to complete nine tests. The experimental result is shown in Table 7.

Through the range analysis of the orthogonal test results, which are shown in Table 8, it can be found that among the process parameters that affect the hydrophobic performance, whether it is the PLA or ABS material, the layer thickness has the largest impact on the hydrophobic performance, followed by the filling method and the printing speed. It can be seen from Figure 12 and Figure 13, which are graphs of the relationship between the index and the influencing factors, that the combination of process parameters with the best hydrophobic performance is L_T3F_M2P_S3. However, this combination of process parameters is not within the scope of the above nine trials. For this reason, it is necessary to conduct experimental verification on the optimal combination of process parameters, and the final measured contact angles are 103.69° (PLA) and 104.60° (ABS).

### 3.3. Discussion of the Experimental Results

(1) 3D printing is an effective processing method that can be used for the preparation of hydrophobic coatings. By adjusting the processing parameters, an ideal surface structure for coating preparation can be obtained. It can be seen from Figure 14 that the obtained surface hydrophobic properties of the test pieces printed under different process parameters by adopting the same coating preparation method are significantly different. According to the above experimental results, it can be seen that the layer thickness and the filling method will significantly affect the surface roughness of the processed test piece so that it will affect the level of the hydrophobic performance. 3D printing can prepare test pieces with a complex structure, which has greater printing flexibility than other traditional processing methods. At the same time, for the processing of single small batch parts, 3D printing has the advantages of fast response speed and low cost. The above characteristics determine the use of 3D printing technology to prepare polymers with hydrophobic properties, and is a low-cost and efficient approach.

(2) As can be seen from the results of this experiment, the maximum contact angle can reach 104.60°. However, we all know that, at present, we can prepare superhydrophobic coatings greater than 150°, and the main reason for the small maximum contact angle in this experiment is that the commercial coating used is a hydrophobic coating, not a superhydrophobic coating. The purpose of this experiment is to verify the feasibility of this method in preparing a hydrophobic coating. If the coating used this time is changed to a superhydrophobic coating, good superhydrophobic properties can also be achieved.

(3) PLA and ABS are used in this experiment, regardless of surface roughness or wettability, and under the same process parameters, the effect of material is not obvious and shows the same trend. However, it should be noted that if the dimensional accuracy and mechanical properties of the prepared parts need to be considered, the rational selection of materials should be considered. These two materials have different forming parameters and different physical and chemical properties. Even under the same process parameters, their dimensional accuracy and mechanical properties are also significantly different.

(4) Because there are many manufacturers of FDM printers, it can be found in the actual printing practice that there are certain differences in the process parameters of different manufacturers. Therefore, when using FDM to prepare the coating, the printing equipment needs to be adjusted.

## 4. Conclusions

For PLA and ABS materials, the surface roughness distribution under different printing process parameters (layer thickness, filling method, and printing speed) was analyzed. Experimental results show that the layer thickness and filling method have a significant effect on the surface roughness. The effect of the filling method on the roughness should be of concern, but the effect of the printing speed on the surface roughness is not obvious.

The orthogonal test method was used to evaluate the wettability. The research results show that the 3D printing method can be used to prepare parts with low cost and a convenient method. The choice of dip coating method can effectively prepare a hydrophobic coating, which significantly improves the hydrophobicity of PLA and ABS performance. At the same time, it can be found that the printing parameters have a very significant effect on the wettability, and it is necessary to pay attention to optimize the selection of the process parameter combination of layer thickness, filling method, and printing speed.

If FDM 3D printing is applied to prepare a hydrophobic coating, the layer thickness and filling method should be optimized because these process parameters significantly affect the surface wettability. In this experiment, the process parameters combination such as the grid filling method and 0.25 mm layer thickness can be used to fabricate the parts with maximum contact angle.

## Figures and Tables

**Figure 1 polymers-12-01539-f001:**
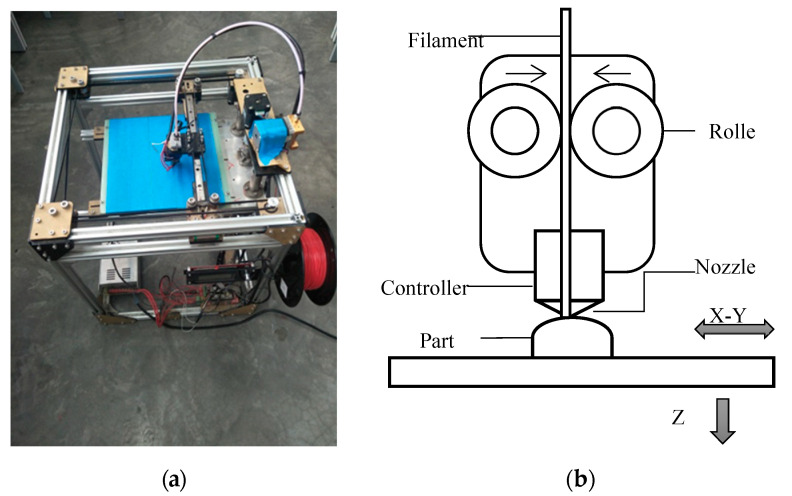
Description of FDM printer: (**a**) The printer applied in this paper; (**b**) the schematic diagram of the FDM process principle.

**Figure 2 polymers-12-01539-f002:**
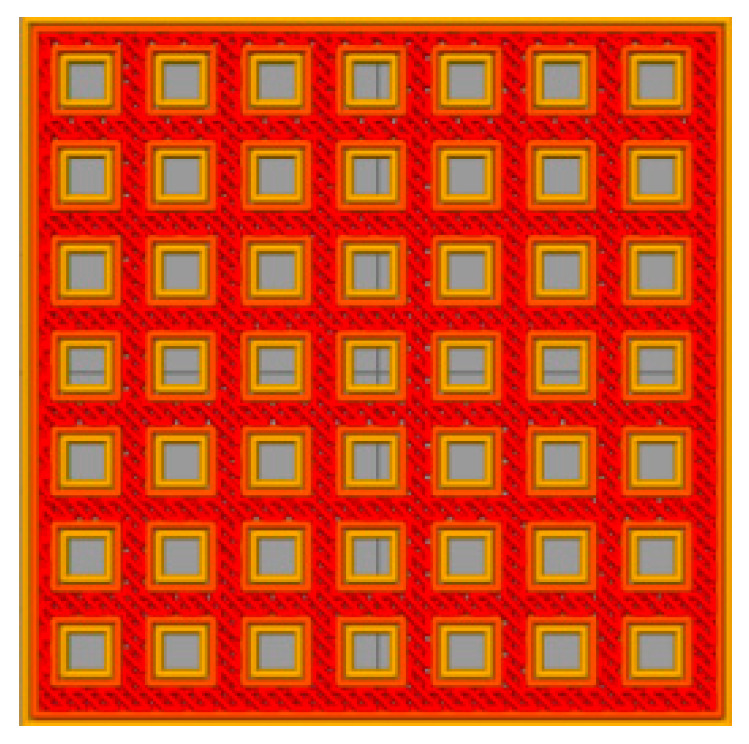
The geometry of samples.

**Figure 3 polymers-12-01539-f003:**
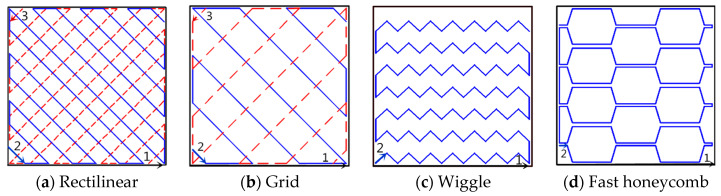
Filling method.

**Figure 4 polymers-12-01539-f004:**
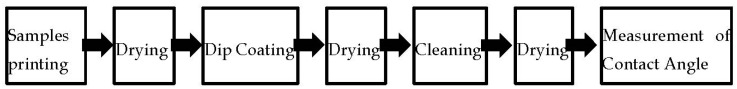
Schematic image of samples preparation.

**Figure 5 polymers-12-01539-f005:**
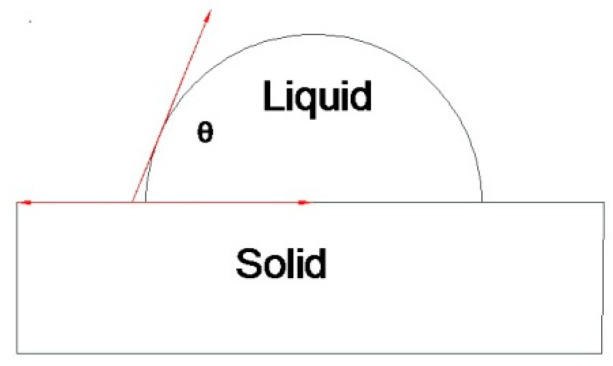
Schematic diagram of contact angle.

**Figure 6 polymers-12-01539-f006:**
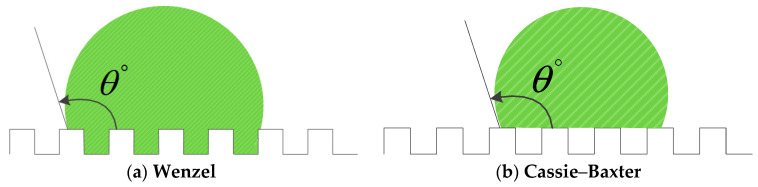
Wenzel model and Cassie model.

**Figure 7 polymers-12-01539-f007:**
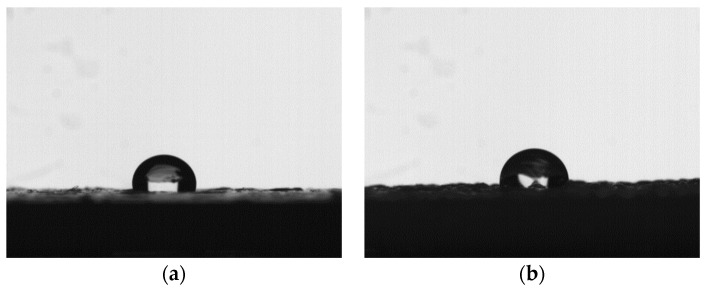
The images of a water droplet on different positions of (**a**) PLA samples without coating, and (**b**) ABS samples without coating.

**Figure 8 polymers-12-01539-f008:**
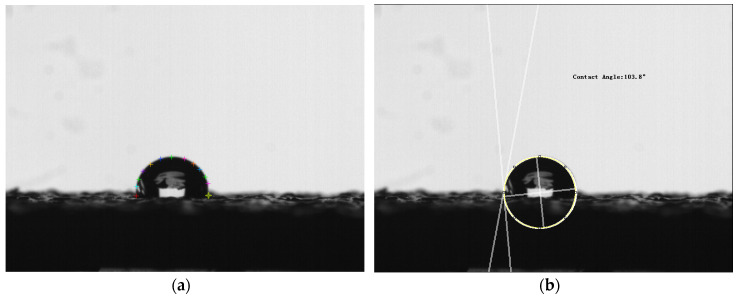
Measurement of contact angle. (**a**) Manual contour extraction; (**b**) measured contact angle.

**Figure 9 polymers-12-01539-f009:**
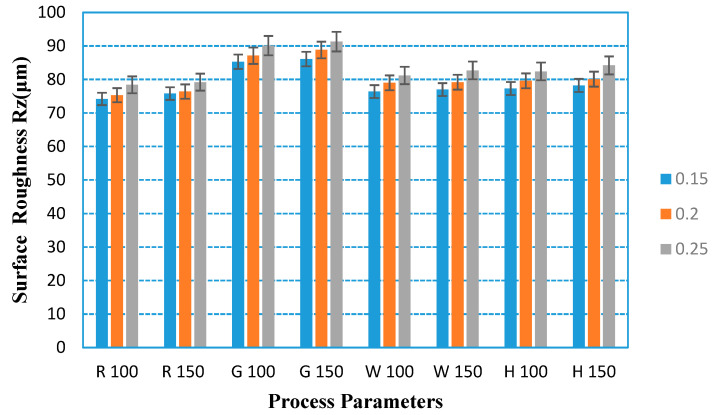
Surface roughness of PLA tested parts under different process parameters.

**Figure 10 polymers-12-01539-f010:**
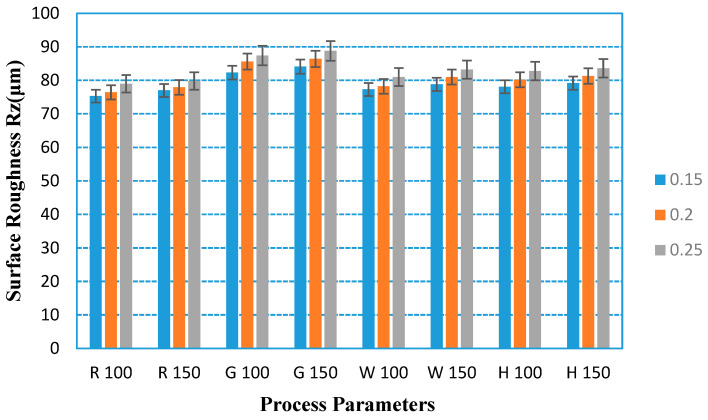
Surface roughness of ABS tested parts under different process parameters.

**Figure 11 polymers-12-01539-f011:**
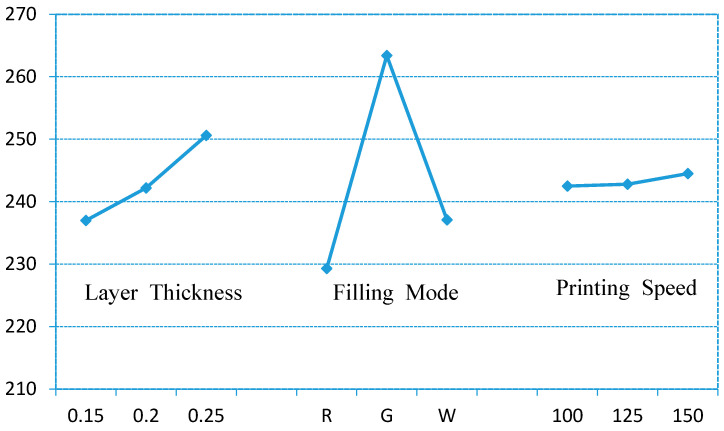
Index-factor.

**Figure 12 polymers-12-01539-f012:**
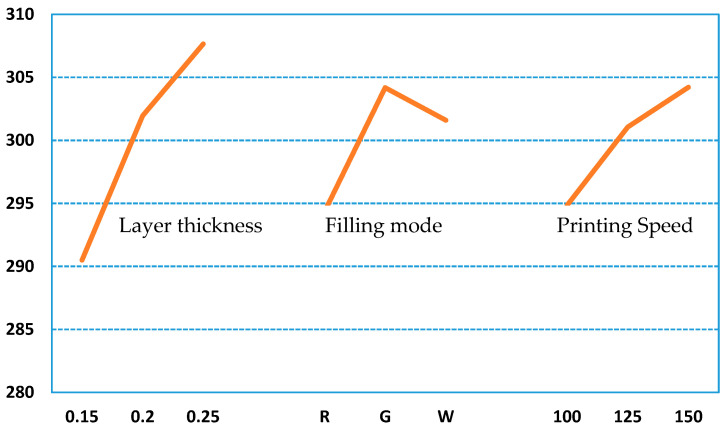
Factor level of contact angle for PLA tested parts.

**Figure 13 polymers-12-01539-f013:**
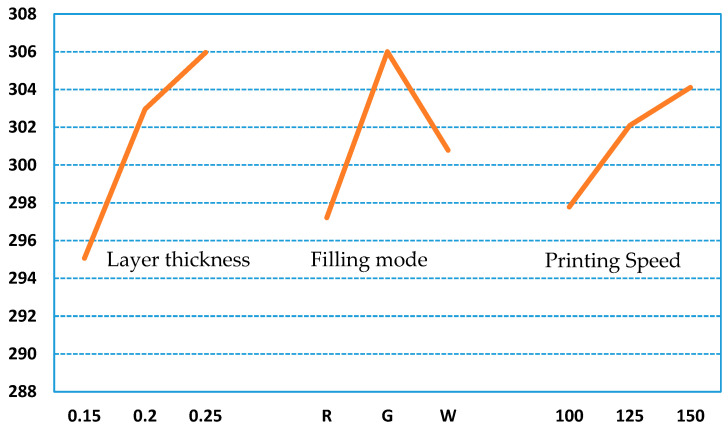
Factor level of contact angle for ABS tested parts.

**Figure 14 polymers-12-01539-f014:**
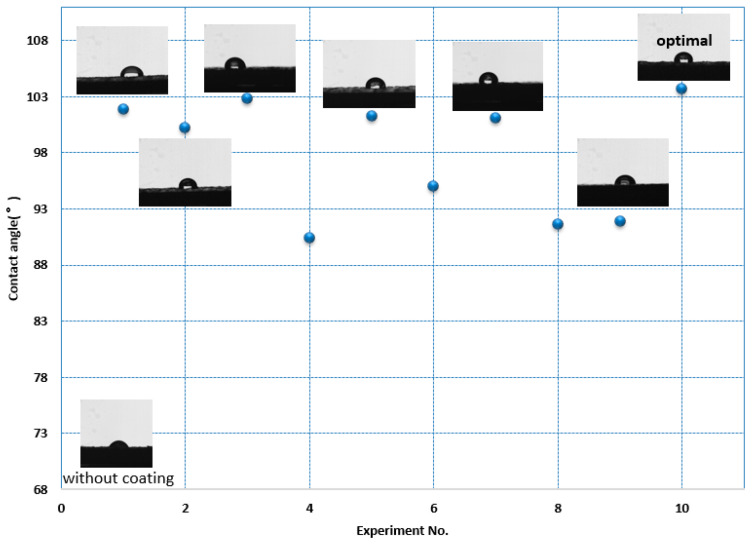
Contact angle of tested parts.

**Table 1 polymers-12-01539-t001:** Characteristics of polylactic acid (PLA) printing materials.

**Brand**	**JGAURORA**	**Color**	**Red**
Material	PLA	Melting Point	190 °C
Diameter	1.75 mm	Print Temperature	190–220 °C
Density	1.25 g/cm^3^	Flexural modulus	≥65 Mpa
Water Absorption	0.5%	Tensile Strength	≥50 Mpa
Hot Bed Temperature	50–60 °C	Flow Rate	5–7 g/10 min

**Table 2 polymers-12-01539-t002:** Characteristics of acrylonitrile butadiene styrene (ABS) printing materials.

**Brand**	**JGAURORA**	**Color**	**Red**
Material	ABS	Melting Point	220 °C
Diameter	1.75 mm	Print Temperature	220–250 °C
Density	1.04 g/cm^3^	Flexural modulus	≥60 Mpa
Water Absorption	1%	Tensile Strength	≥43 Mpa
Hot Bed Temperature	50–60 °C	Flow Rate	2–4 g/10 min

**Table 3 polymers-12-01539-t003:** Measurement of light-section microscope (unit: mm).

$a_{1}$	$a_{2}$	$a_{3}$	$a_{4}$	$a_{5}$	$a_{6}$	$a_{7}$	$a_{8}$	$a_{9}$	$a_{10}$
3.8	3.68	3.88	3.74	3.85	3.7	3.9	3.78	3.85	3.75

**Table 4 polymers-12-01539-t004:** Surface roughness of PLA and ABS tested parts.

Process Parameters				Surface Roughness Rz (μm)	
No.	LT(mm)	FM	PS(mm/s)	PLA	ABS
1	0.15	R	100	74.2	75.3
2	0.15	G	100	85.3	82.3
3	0.15	W	100	76.4	77.3
4	0.15	F	100	77.3	78.1
5	0.20	R	100	75.3	76.4
6	0.20	G	100	87.1	85.6
7	0.20	W	100	79.0	78.2
8	0.20	F	100	79.6	80.2
9	0.25	R	100	78.4	79.0
10	0.25	G	100	90.1	87.4
11	0.25	W	100	81.2	81.0
12	0.25	F	100	82.4	82.8
13	0.15	R	150	75.8	77.0
14	0.15	G	150	86.1	84.1
15	0.15	W	150	77.0	78.8
16	0.15	F	150	78.2	79.2
17	0.20	R	150	76.4	77.9
18	0.20	G	150	88.8	86.4
19	0.20	W	150	79.2	81.0
20	0.20	F	150	80.1	81.3
21	0.25	R	150	79.2	79.8
22	0.25	G	150	91.3	88.8
23	0.25	W	150	82.7	83.2
24	0.25	F	150	84.2	83.6

**Table 5 polymers-12-01539-t005:** Orthogonal test table of surface roughness measurement.

Layer Thickness (μm)	Filling Mode	Print Speed (mm/s)	Surface Roughness R_Z_ (μm)
3 (0.25)	3 (W)	1 (100)	81.2
1 (0.15)	2 (G)	3 (150)	86.1
3	1 (R)	3	79.2
1	3	2 (125)	76.7
2 (0.20)	3	3	79.2
3	2	2	90.2
2	2	1	87.1
2	1	2	75.9
1	1	1	74.2

**Table 6 polymers-12-01539-t006:** Range analysis of surface roughness.

Range	L_T	F_M	P_S
K1	237	229.3	242.5
K2	242.2	263.4	242.8
K3	250.6	237.1	244.5
k1	79	76.43	80.83
k2	80.73	87.8	80.93
k3	83.53	79.03	81.5
R	4.53	11.36	0.67

**Table 7 polymers-12-01539-t007:** Experimental result of contact angle.

Layer Thickness (μm)	Filling Mode	Print Speed (mm/s)	Contact Angle	
			PLA	ABS
3 (0.25)	3 (W)	1 (100)	101.90	100.25
1 (0.15)	2 (G)	3 (150)	100.17	100.47
3	1 (R)	3	102.78	101.91
1	3	2 (125)	98.43	98.8
2 (0.20)	3	3	101.27	101.73
3	2	2	102.98	103.8
2	2	1	101.04	101.73
2	1	2	99.65	99.5
1	1	1	91.91	95.8

**Table 8 polymers-12-01539-t008:** Range analysis of contact angle measurement.

Range	PLA			ABS		
	L_T	F_M	P_S	L_T	F_M	P_S
K1	290.51	294.34	294.85	295.07	297.21	297.78
K2	301.96	304.19	301.06	302.96	306	302.1
K3	307.66	301.60	304.22	305.96	300.78	304.11
k1	96.84	98.11	98.28	98.36	99.07	99.26
k2	100.65	101.40	100.35	100.99	102	100.7
k3	102.55	100.53	101.41	101.99	100.26	101.37
R	5.72	3.28	3.12	3.63	2.93	2.11

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
