# Peer review of "Preparation of Hydrophobic Surface on PLA and ABS by Fused Deposition Modeling"

_polymers, 2020, doi:10.3390/polym12071539_

Round 1

Reviewer 1 Report

The study deals with the FDM 3D printing technology, the design and the processing flexibility were applied to the preparation of hydrophobic coatings on PLA and ABS parts.

The paper is interesting and well organized. It is also of secure interest for the community of the journal and it should be improved by adding more references in the bibliography. The conclusions should be also improved by pointing out the main results of the study. 

Singh, et al. "Multi-material additive manufacturing of sustainable innovative materials and structures." Polymers 11.1 (2019): 62.

Zander, N. E. (2019). Recycled polymer feedstocks for material extrusion additive manufacturing. In Polymer-Based Additive Manufacturing: Recent Developments (pp. 37-51). American Chemical Society.

Author Response

Response to Reviewer 1 Comments

Point 1: The paper is interesting and well organized. It is also of secure interest for the community of the journal and it should be improved by adding more references in the bibliography. The conclusions should be also improved by pointing out the main results of the study. 

Response 1: Thanks for your suggestion. Now, I have revised the manuscript. In the introduction of revised manuscript, the recent advancements in related fields (FDM printing technology, hydrophobic preparation method and 3D printing process optimization) have been added. In the conclusion, one paragraph has been added to point out the main result. Thanks again.

Reviewer 2 Report

The manuscript polymers-846586 describes 3D printing of plastic samples of PLA and ABS, which were subsequently coated with a nano-structured hydrophobic coating surface. The authors correlate the measurement of the contact angle with the printing process parameters and the surface roughness. And they assure that experimental results show that the layer thickness and filling method have a significant effect on the surface roughness of 3D printed parts, while the printing speed has no effect on the surface roughness.

The report can be interesting for the research groups that use 3D printing technology as a technique of processing their samples. However, this reviewer is unsure whether this manuscript is appropriate for the journal polymers.

If the polymer journal editor considers this report to be in line with the journal's objectives and scope, then, before accepting it, I recommend that the authors address the following comments and questions.

1)- The objective of the investigation must be written clearly and directly in the abstract and in the introduction.
1.1- To whom will the data and conclusions contained in this report be useful?
1.2- Does this report provide any technical basis for other future research using 3D printing as a thermoforming technique?

In section 3-Results and Discussion.

2)- It is mentioned that the tests were carried out in triplicate. However, in figures 9 and 10 the bars that show the variance of the average surface roughness do not show a Statistical significance (statistically significant variation).
2.1- With three data per specimen, is it possible to affirm that the results in the data are not explainable by chance alone?
2.2- What is the number of events that statistical theory requires to ensure the Gaussian behaviour of the data?

In section 3-Results and Discussion.

3)- Figures 11, 12, and 13 are used to conduct a discussion and analysis on the relationship between the 3D printed's parameters and hydrophobic properties. However, there are only three experimental points in each graphic profile.
3.1- Is it possible to obtain a forceful trend with only three experimental points?
3.2- Can 2- or 3-degree variations in contact angle values be attributable to software or instrumental issues?

Author Response

Thanks for your suggestions.

1)- The objective of the investigation must be written clearly and directly in the abstract and in the introduction.
1.1- To whom will the data and conclusions contained in this report be useful?

Response 1.1: Thanks for your suggestion. It is very important to introduce the objective of the investigation. I have revised the original manuscript. In the second paragraph of the introduction, the occasions where the method and technology can be applied in the future are supplemented.

1.2- Does this report provide any technical basis for other future research using 3D printing as a thermoforming technique?

Response 1.2: Thanks for your suggestion. 3D printing technology, especially for polymer FDM3D printers, has found limitations in printing accuracy and size in previous research, which restricts the further industrial application of this technology. However, recent research has focused on large-scale industrial-grade FDM 3D printing technology and equipment, and has made some progress. It may be more widely used in many industries in the future. This paper only focuses on changing the properties of the 3D printed surface. This requirement has a very broad future in environments that are resistant to corrosion, ice, and oil. The printing technology is relatively mature, so the application of this method in industry has a solid theoretical foundation.

In section 3-Results and Discussion.

2)- It is mentioned that the tests were carried out in triplicate. However, in figures 9 and 10 the bars that show the variance of the average surface roughness do not show a Statistical significance (statistically significant variation).
2.1- With three data per specimen, is it possible to affirm that the results in the data are not explainable by chance alone?

Response 2.1: Thanks for your suggestion.For the experiment of surface roughness, we selected three process parameters of layer thickness, filling method, and printing speed, a total of 24 combinations for experimental analysis. For each parameter combination, we prepared 3 test pieces, a total of 72 test pieces were prepared. In order to eliminate accidental random errors, three test pieces are printed at the same time, which can ensure that the printing device is completed in a unified state, and can ensure the consistency of the printing parameters. After that, we also conducted orthogonal test analysis, and the results also proved that there is no random error.

2.2- What is the number of events that statistical theory requires to ensure the Gaussian behaviour of the data?

Response 2.1: Thanks for your suggestion.For Figure 9 and Figure 10 you mentioned, it is based on the average and standard deviation obtained from 72 experimental results. The horizontal axis of the graph represents 24 process parameter combinations. The experimental process is independent of each other, so the overall graph trend is The actual test results reflect that there is no need to satisfy the normal distribution between them.

In section 3-Results and Discussion.

3)- Figures 11, 12, and 13 are used to conduct a discussion and analysis on the relationship between the 3D printed's parameters and hydrophobic properties. However, there are only three experimental points in each graphic profile.
3.1- Is it possible to obtain a forceful trend with only three experimental points?

Response 3.1: Thanks for your suggestion.The experimental data in Figures 11, 12 and 13 are not only from three experimental points, but from the experimental results of the orthogonal test, which were obtained by factor level analysis. The main purpose is to compare various influencing factors, so as to draw significant influencing factors

3.2- Can 2- or 3-degree variations in contact angle values be attributable to software or instrumental issues?

Response 3.2: Thanks for your suggestion. The software and measurement equipment you mentioned will indeed affect the measured value of the contact angle, but the measurement and software analysis error will not reach 2 degrees. If the measurement error reaches 2 degrees, this measuring instrument cannot be used. The measurement equipment we used in this experiment has very high accuracy, so the measurement results are credible.

Round 2

Reviewer 2 Report

I agree with the authors' responses to my comments.

I suggest doing an English style check to find fine/minor spell corrections.